# The Microtubule-Associated Innate Immune Sensor GEF-H1 Does Not Influence Mouse Norovirus Replication in Murine Macrophages

**DOI:** 10.3390/v11010047

**Published:** 2019-01-10

**Authors:** Svenja Fritzlar, Peter A. White, Jason M. Mackenzie

**Affiliations:** 1Department of Microbiology and Immunology, University of Melbourne at the Peter Doherty Institute for Infection and Immunity, Melbourne, VIC 3010, Australia; svenja.fritzlar@monash.edu; 2School of Biotechnology and Biomolecular Sciences, The University of New South Wales, Sydney, NSW 2052, Australia; p.white@unsw.edu.au

**Keywords:** mouse norovirus, innate immunity, microtubules, GEF-H1

## Abstract

Norovirus is an acute infection of the gastrointestinal tract causing rapid induction of vomiting and diarrhoea. The infection is sensed and controlled by the innate immune system, particularly by the RNA helicase MDA-5 and type I and III interferons (IFNs). We have observed that intracellular replication of murine norovirus (MNV) occurs in membranous clusters proximal to the microtubule organising centre, a localisation dependent on intact microtubules. Recently, it was shown that the host protein guanine nucleotide exchange factor-H1 (GEF-H1) is a microtubule-associated innate immune sensor that activates interferon Regulatory Factor 3 to induce the production of type I IFNs. Thus, we interrogated the potential role of GEF-H1 in controlling MNV infections. We observed that GEF-H1 was recruited to the MNV replication complex; however RNAi-mediated suppression of GEF-H1 did not outwardly affect replication. We furthered our studies to investigate the impact of GEF-H1 on MNV innate detection and observed that GEF-H1 did not contribute to type I IFN induction during MNV infection or influenza virus infection but did result in a small reduction of interferon–β (IFNβ) during West Nile virus infection. Intriguingly, we discovered an interaction of GEF-H1 with the viral MNV non-structural protein 3 (NS3), an interaction that altered the location of GEF-H1 within the cell and prevented the formation of GEF-H1-induced microtubule fibres. Thus, our results indicate that GEF-H1 does not contribute significantly to the innate immune sensing of MNV, although its function may be modulated via interaction with the viral NS3 protein.

## 1. Introduction

Norovirus (NoV) infects approximately 700 million people each year resulting in an estimated 220,000 deaths in developing countries [1,2,3] and an economic burden of USD $60 billion due to healthcare costs and productivity loss [4,5]. NoVs are positive sense single-stranded RNA (ssRNA) viruses within the *Calicivirdae* family and are classified into seven genogroups [6,7,8], based on genetic diversity of the viral VP1 (or capsid) [6]. Unfortunately, little is known about human NoV replication and pathogenesis due to the difficulty of cultivating the virus in the laboratory. However, it was recently observed that human B cells exposed to enteric bacteria or enteroid cultures were permissive and susceptible to human NoV infection [9,10]. Until this point the only norovirus to be studied in culture was the recently discovered genogroup V murine norovirus (MNV) [11,12]. MNV is a natural pathogen of mice and displays a tropism for dendritic cells and macrophages, with the innate immune system playing a major role in the detection and clearance of the virus [12,13,14].

Our laboratory has been instrumental in determining the intracellular replication of MNV and its interactions with cellular membranes and components [15,16,17,18]. We previously investigated the interaction of MNV with the cytoskeleton of the host cell and observed that tubulin, a major component of microtubules, co-localises with the MNV replication complex (RC) and the MNV non-structural protein 3 (NS3) [15,16]. Microtubules are highly dynamic structures, which can undergo fast and drastic changes through the depolymerisation and polymerisation at the end of the tubular fibres. Microtubules have several functions within cells, including the motility and structure of the cell, as well as providing a scaffold for the intracellular transport of proteins and vesicles. Experiments using Nocodazole, a drug that leads to the depolymerisation of microtubules, showed that a functional microtubule network was needed for the successful generation of a concentrated MNV RC during virus replication [16]. This indicates that MNV may use the host microtubule network for the transport of host and viral proteins and benefits from the structural scaffold to set up an efficient replication complex.

Recently, we reported a direct interaction between β–tubulin and vesicular structures induced upon transient expression of the MNV non-structural protein NS3 [15]. The motility and size of the NS3-induced vesicles were dependent on a functional microtubule network, implying that NS3 might contribute to the microtubule-dependent formation of the MNV RC. Additionally, MNV not only seems to be dependent on the microtubule network but also causes changes to it. As early as 12 h after an infection with MNV, the formation of strongly polymerising microtubule bundles in the cells periphery were observed [16].

Intriguingly, a similar phenotype of microtubule bundling has been observed in cells which express guanine nucleotide exchange factor-H1 (GEF-H1), which, unlike other GEFs, is able to bind microtubules [19,20]. GEFs are regulators of GTPases, activating the GTPase by mediating the exchange from GDP to GTP. GEF-H1 has been shown to have several regulatory functions in the cell through its ability to activate the GTPases Rho and Rac [21,22,23,24]. It is involved in the connection of microtubules with the actin cytoskeleton, in the regulation of tight junctions, and in vesicle transport [20,23]. During influenza A virus (IAV) and Mycobacterium tuberculosis (M.tb) infections, GEF-H1 has been proposed to play a major role in the innate immune response and the activation of transcription factors which lead to the expression of interferon–β (IFNβ) messages [25,26]. The authors proposed a model whereby GEF-H1 is activated upon detection of pathogen-associated molecular patterns to induce inflammatory cytokine and IFNβ production to promote pathogen elimination.

Due to the involvement of GEF-H1 in the innate immune response against IAV and M.tb and to its interaction with the microtubule network, we were interested in the role of GEF-H1 during MNV infection and replication. Considering its interaction with the microtubule network and the recognition of intracellular pathogens, GEF-H1 could be an important factor in the host defence against MNV or a target for MNV proteins.

## 2. Materials and Methods

### 2.1. Cell Lines and Virus Infection

RAW264.7 (a murine macrophage cell line), Vero, and HEK-293T cells were purchased from ATCC (Manassas, VA, USA) and maintained in Dulbecco’s modified Eagle’s medium (DMEM) with 10% Foetal bovine serum (FBS) and 2 mM GlutaMAX™ (all Gibco, Waltham, MA, USA). All cell lines were cultivated at 37 °C and 5% CO_2_. RAW264.7 macrophages were infected with a multiplicity of infection (MOI) of 5 with a tertiary stock of the MNV strain CW1 [12]. The culture medium was replaced with a serum-free medium with GlutaMAX™. The virus was added, and the culture dishes were rocked regularly for about 60 min to ensure virus binding and to prevent the drying of cells. Afterwards, a sufficient amount of the serum-free medium was added, depending on the culture vessel. If not indicated differently, the cells were harvested or fixed 12 h after infection.

### 2.2. Plasmids and Antibodies

Plasmids encoding the 6× HIS-tagged MNV non-structural (NS) proteins (NS1-2–NS7) on a pcDNA3.1 backbone were generated and published previously [17,27]. The GEF-H1 cDNA expression plasmids were kindly donated by Hans–Christian Reinecker [25]. The antibodies to the following antigens were used: anti-6× HIS and anti-calnexin (Abcam, Cambridge, MA, USA), anti-actin (Sigma, St. Louis, MO, USA), anti-dsRNA (SCISONS, Budapest, Hungary), anti-GEF-H1 (Pierce, Puyallup, WA, USA), anti-β–tubulin (Molecular Probes, Eugene, OR, USA), MNV NS3 and NS6 (kindly provided by Kim Green, NIH, USA), MNV NS5 and NS7 (manufactured by Invitrogen, Carlsbad, CA, USA).

### 2.3. Transfection of Cell Lines

Vero and HEK-293T cells were transfected with Lipofectamine^®^2000, and cells were treated according to the manufacturer’s protocol. In brief, plasmid DNA as well as the transfection reagent were diluted in Opti-MEM^®^ media, mixed together, and incubated at room temperature. Plasmid DNA and transfection reagent mixes were added dropwise to the cells kept in DMEM with 10% FBS and 2 mM GlutaMAX™ (all Gibco, Thermo Fisher Scientific, Waltham, MA, USA). Cells were incubated at 37 °C for 18 h before analysis.

### 2.4. Immunofluorescence Staining and Confocal Imaging

Cells intended for confocal imaging analysis were grown on 10 mm round coverslips and treated according to the experimental design as described previously [18]. Briefly, cells were fixed in 4% (*w*/*v*) paraformaldehyde (PFA) (Electron Microscopy Sciences, Hatfield, England) (in PBS) and permeabilised in 0.1% (*w*/*v*) Triton™ X-100 (Sigma, St. Louis, MO, USA) and 4% PFA, stained with the primary antibody in 1% Bovine serum albumin (BSA in PBS. After washing the cells in 0.1% BSA (in PBS), the secondary antibody (species-specific Alexa Fluor antibody, Life Technologies, Carlsbad, CA, USA) was added and then washed in PBS and 4′,6-diamidino-2-phenylindole (DAPI; 0.5 µg/mL) was added for 5 min. Final washing steps were performed with Milli-Q water (Millipore, Burlington, MA, USA), and coverslips were dried and mounted on cover slides with Ultramount #4 (Fronine, Riverstone, NSW, Australia). Samples used for staining with the GEF-H1 antibody were fixed in ice-cold methanol for 3 min followed by a 30 s incubation with ice-cold acetone. Samples were kept cool and dark until imaging with confocal microscopes. Confocal pictures were collected either with the LSM 700 or LSM 710 confocal microscope using the ZEN^®^ software (Zeiss, Oberkochen, Germany). The same conditions were used when capturing images from these experiments.

### 2.5. Immunoblotting

Samples were lysed in a NP40 lysis buffer (150 mM NaCl, 50 mM Tris pH 8.0, 1% NP-40) and centrifuged at high speed for 10 min to sediment cell debris. All protein samples were handled at 4 °C or on ice and supplied with the Protease Inhibitor Cocktail III (Astral Scientific, Taren Point, NSW, Australia). Protein samples were boiled at 95 °C in a SDS loading buffer (125 mM Tris-HCl, 4% SDS, 20% glycerol, 10% 2-mercaptoethanol, 0.004% bromophenol blue), separated on a polyacrylamide gel (10% or 12% depending on the size of the protein of interest) using SDS-PAGE (120 V), and then transferred onto a 0.2 µm Polyvinylidene difluoride (PVDF) membrane (Bio-Rad, Laboratories, Inc., Hercules, CA, USA) (100 V, 65 min). Membranes were processed according to manufacturer’s protocol and incubated in 5% BSA (solved in 0.1% Tween in PBS) at 4 °C overnight or at RT for at least 2 h. Primary antibodies were diluted in 5% BSA and 0.1% Tween in PBS and incubated with the membrane for 3–4 h at room temperature or at 4 °C overnight. After the wash steps with 0.1% Tween in PBS, the secondary antibodies in the 0.1% Tween in PBS were added to the membrane (2–3 h) before they were incubated with an Enhanced Chemiluminescence (ECL) Plus Western Blotting Substrate (Pierce, Thermo Fisher, Waltham, MA, USA) for 5 min and signals were visualised with MF-ChemiBis documentation system (DNR, Jerusalem, Israel).

### 2.6. Quantitative RT-PCR

Cells were harvested and washed in PBS before cell pellets were lysed in TRIzol^®^ Reagent (Life Technologies, Carlsbad, CA, USA) for 10 min and stored at −80 °C. The RNA was extracted adding chloroform in a ratio of 1:5, mixing thoroughly and incubating the mixture for 5 min at RT to allow separation of the aqueous and organic phase. After centrifugation at 13,500 *g* for 15 min at 4 °C, the organic phase was discarded and 20 ng glycogen was added to the aqueous phase. RNA was precipitated in isopropanol (1:2 ratio to amount of TRIzol^®^) for 10 min and centrifuged at 13,500 *g* for 10 min at 4 °C. The supernatant was removed, and the RNA pellet was washed with 70% ethanol, centrifuged again for 5 min, and after removal of the ethanol, left at RT to dry. RNA pellets were dissolved in autoclaved DEPC-treated water at 60 °C for 10 min. For consecutive reverse transcription, the RNA concentration was measured using NanoDrop™ (Thermo Scientific, Waltham, MA, USA). To exclude DNA contamination, 1 µg RNA was treated with RQ1 DNase (Promega, Madison, WI, USA) and incubated for 30 min at 37 °C. DNase was heat inactivated at 65 °C for 15 min with EDTA. For reverse transcription, Sensifast RT (Bioline, Alexandria, NSW, Australia) was used and the following incubation pattern was applied: 25 °C for 10 min, 42 °C for 15 min, and 85 °C for 5 min. The gene-specific primers and 2× ITaq Universal Sybr Green Supermix (Bio-Rad, Hercules, CA, USA) were used to set up the qPCR samples in duplicates (primer sequences and cycling programs can be requested from the authors) as previously described [16]. RNA levels were analysed with a Stratagene Mx3005P™ qPCR machine. The fold change in the target genes was calculated using the ΔΔCT method relative to the internal control *GAPDH* gene.

### 2.7. Plaque Assay

To identify viral titres, plaque assays were performed on RAW264.7 cells as previously described [18]. Briefly, cells were seeded in 6-well plates 24 h before the assay was performed to reach 50% confluency the next day. A serial dilution (1:10) of the viral samples was performed in DMEM and the 10^−3^ to 10^−8^ dilutions were tested in the plaque assay. The media was removed from the cells and replaced with the serial dilutions in duplicates. Plates were incubated for 1 h at 37 °C with regular rocking to ensure viral binding and to avoid the drying of the monolayer. Next, an LMP agar overlay (9.4 mL 1 × DMEM, 0.33 mL FBS, 0.2 mL 0.9 M NaHCO3, 0.33 mL 1 M HEPES, 0.12 mL Glutamax, 3.1 mL 1.5% LMP agarose for one 6-well plate) was added to the wells and set at 4 °C for 30 mins. Afterwards, plates were incubated at 37 °C for 48 h before fixing the cells with 10% formalin for 1 h. To visualise occurring plaques, cells were stained with 0.2% crystal violet (in 10% methanol/PBS).

### 2.8. Immunoprecipitation

Transfected with either pcDNA3.1-NS3-HIS or RG204546-GEF-H1-GFP or co-transfected with both plasmids, 293T cells were lysed in a NP40 buffer (150 mM NaCl, 50 mM Tris pH 8.0, 1% NP-40). Cell debris was discarded, and lysates were incubated with cOmplete^TM^ His-Tag Purification Resin (Roche, Basel, Switzerland) for 1 h at RT or at 4 °C overnight. The Lysate and resin mixes were processed according to the manufacturer’s protocol, resulting in fractionation of His resin-bound proteins and residual cellular proteins.

### 2.9. GEF-H1 Silencing Via Sirna Treatment

Three siRNAs (siRNA1-3) against mouse GEF-H1 mRNA were purchased from Bioneer Pacific and tested for their efficiency to knock down the GEF-H1 expression separately, as well as in combination. Only siRNA3 showed a significant reduction in the GEF-H1 expression and was used in all the following experiments. Lipofectamine^®^RNAiMAX was used to transfect RAW264.7 cells with siRNA according to the manufacturer’s protocol. In brief, 40 pmole siRNA was diluted in Opti-MEM^®^ as well as Lipofectamine^®^RNAiMAX. The diluted Lipofectamine^®^ and siRNA were mixed and incubated for 5 min at room temperature. The cell culture media was exchanged and the siRNA and Lipofectamine^®^ mixture were added dropwise to cells (12-well). Cells were incubated at 37 °C for 24 h before they were treated again with 40 pmole siRNA. If cells were infected after the siRNA knockdown, the virus was added to the cells 12 h after the second siRNA treatment. For cells which were treated twice with the siRNA 24 h apart, cells were seeded at a lower confluency of 30–50% at the time of the first siRNA treatment to avoid overgrowth.

## 3. Results

### 3.1. GEF-H1 Is Found within the MNV Replication Complex

Previous observations have shown that infection with MNV leads to changes in the cytoskeleton of infected cells [16]. The most obvious effect observed was in the microtubule network with the induction of microtubule bundles. Thus, we aimed to investigate the function of GEF-H1 in MNV replication and analysed the location and distribution of GEF-H1 during infection. Mouse macrophages were infected for 12 h and 18 h (MOI 5), fixed, and subsequently immune-stained with anti-GEF-H1 and anti-NS4 antibodies (Figure 1). Uninfected cells displayed a cytoplasmic and microtubule-like staining pattern for GEF-H1 (Figure 1A panels a–d), indicating the association of GEF-H1 with microtubules, but we did not observe thick fibre structures indicative of microtubule bundling. The MNV-infected cells displayed only minor changes in the morphology of GEF-H1 at 12 h (Figure 1A panels e–h) or 18 h (Figure 1A panels i–l) after infection without a clear induction of microtubule fibres. However, we did observe that some of the cytoplasmic GEF-H1 co-localised with the MNV RC in the perinuclear region at both time points analysed.

These results suggest that GEF-H1 might be recruited or sequestered within the MNV RC. This redistribution could be attributed to either (i) a requirement for MNV replication, (ii) a host response involved in the sensing of MNV and the activation of the antiviral immune response, or (iii) targeting by the virus to avoid immune detection and to dampen the anti-viral response.

### 3.2. Expression of GEF-H1 Leads to Changes in the Localisation of the RC during MNV Infection

To interrogate the role of GEF-H1 during a MNV infection, we utilised recombinant cDNA plasmids that encoded wildtype GEF-H1 tagged to GFP or mutant GEF-H1 constructs [25]. C53R and ∆DH are loss of function mutants, lacking the ability to associate with the microtubules or missing the catalytic function, respectively, whereas S885A is a constitutively active mutant, which can still bind to the microtubules. Murine macrophages were transfected with the different GFP-tagged GEF-H1 encoding plasmids and subsequently infected with MNV. Cells were fixed at 12 h.p.i. and stained with antibodies against the viral protein NS4, as well as the nuclear stain DAPI, and analysed with immunofluorescence microscopy (Figure 2).

The expression of the wildtype GEF-H1, ∆DH, and S885A mutants lead to significant changes in the morphology and location of the MNV RC (Figure 2 panels a–c and i–o). We observed that instead of one large perinuclear-located RC, several smaller RCs could be observed distributed throughout the infected cell. These RCs were distributed in a non-polar fashion and generally observed around the cell periphery. In contrast, cells transfected with the C53R mutant (lack of microtubule binding), and subsequently infected, displayed no change in the morphology or location of the MNV RC, which is located in the perinuclear region and forms a single aggregate (Figure 2 panels e–g). This would suggest that the above interference of the MNV RC formation is dependent on the ability of GEF-H1 to bind microtubules.

### 3.3. The MNV Protein NS3 Co-Localises with GEF-H1 and Alters Its Distribution

To elucidate the effect of GEF-H1 on MNV, or the virus on GEF-H1, we co-expressed the viral NS proteins and the major capsid protein VP1 together with the wildtype GFP-tagged GEF-H1 in Vero cells. The cells were fixed at 24 h after transfection and stained with antibodies against the 6× HIS tag of the viral proteins for immunofluorescence analysis (Figure 3A). In general, the co-expression of GEF-H1 with the MNV proteins did not affect the localisation of GEF-H1 or lead to co-localisation nor did GEF-H1 have an influence on the viral protein localisation (only NS1-2 and NS4 are shown in Figure 3A, others in Appendix A). Intriguingly though, the co-expression of MNV NS3 and GEF-H1 altered the localisation and appearance of GEF-H1 as well as NS3 (Figure 3A panels e–h, see Appendix A for images of individual expression of NS3). Upon co-expression with NS3, the GEF-H1 expression did not induce microtubule bundling but was instead observed to be evenly distributed throughout the cell and the cell periphery, except for the nucleus and vesicle-like structures. We additionally observed that the NS3 staining mainly co-localised with the GEF-H1 and was also observed to have a more diffuse cytoplasmic staining compared to the expression of NS3 alone [15,16].

To further validate the observed co-localisation of NS3 and GEF-H1 in the immunofluorescence analysis, both proteins were co-expressed in the 293T cells and analysed via coprecipitation to determine if these two proteins interacted directly. The expressed NS3 protein was precipitated by utilizing its 6× HIS tag and a HIS bead-based assay. The precipitate was subsequently analysed by immunoblotting and examined for the presence of NS3 (anti-6× HIS antibody) and GEF-H1 (anti-GEF-H1 antibody) (Figure 3B). A 6× HIS positive signal corresponding to the size of NS3 (39 kDa) was detected in the precipitate of the lysate of NS3 only and NS3 and GEF-H1 expressing cells, confirming the successful precipitation of NS3. GEF-H1 could be detected in the lysate of the GEF-H1 only and GEF-H1 and NS3 transfected cells, verifying the successful expression and co-expression of the protein. Analysing the NS3 precipitate, GEF-H1 could be detected in cell lysates co-expressing NS3. This indicates that GEF-H1 not only co-localises with NS3 but also can bind to it.

After observing that GEF-H1 co-localised and interacted directly with MNV NS3, we were interested in investigating the interaction of NS3 with the GEF-H1 mutants. For this, Vero cells were used to co-express NS3 with the three different GEF-H1 mutants C53R, ∆DH, and S885A. Cells were stained with the anti-6× HIS antibody and analysed using immunofluorescence microscopy (Figure 4). Surprisingly, the cellular distribution of all mutants seemed to be affected when cells co-expressed MNV NS3. This included the C53R mutant which lacks the ability to bind to the microtubules and did not have an effect on the location and size of the MNV RC. Comparable to the co-expression of NS3 and the wildtype GEF-H1, the morphology of the mutants was dispersed and diffused (Figure 4 panels e–p). Co-expression of NS3 and the constitutively active mutant S885A appeared to have the lowest changes in morphology, including the formation of microtubule fibres (Figure 4 panels m–p). Those fibres were less thick and prominent than previously observed in the single expression of the protein but were still inducible in contrast to the wildtype GEF-H1.

These observations would indicate that MNV NS3 is affecting the capacity of GEF-H1 to interact with microtubules, perhaps via direct binding, and that this interaction may perturb the activity and function of GEF-H1 during the MNV replication cycle.

### 3.4. siRNA Knockdown of Endogenous GEF-H1 Does Not Affect MNV Replication Complex Formation or Localisation

To disentangle the interaction of GEF-H1 and MNV and to further investigate the function of GEF-H1 during MNV infections, we used siRNA to suppress the expression of GEF-H1 in murine macrophages. Cells were treated twice with siRNA specific for GEF-H1 to ensure efficient knockdown of the protein before they were infected with MNV (MOI 5) for 12 h. The successful knockdown of GEF-H1 was verified via immunoblotting and immunofluorescence (Figure 5A,B). Macrophages treated with GEF-H1 siRNA had a significantly lower signal for endogenous GEF-H1 compared to cells that were treated with the control siRNA or left untreated. Even though the knockdown was robust in the immunoblot analysis, our IFA revealed a few cells still expressing GEF-H1 at a level comparable to the control cells in the control siRNA-treated cells (Figure 5B). If the number of these cells was less than 5%, the knockdown was considered successful.

Staining for the viral protein NS4 revealed that there was no significant difference in the number of infected cells or the shape and location of the MNV RC in GEF-H1 siRNA-treated cells or control treated cells (Figure 5B). These findings suggested that GEF-H1 was not essential or required for formation or positioning of the MNV RC.

To extend the previous observation that GEF-H1 does not influence the formation or localisation of the viral replication complex, we determined if GEF-H1 played any other role during the MNV replication cycle. Thus, we determined the amount of viral mRNA and enumerated the production of infectious virus produced during replication in GEF-H1 silenced cells compared to untreated cells (Figure 5C,D, respectively). For this, murine macrophages were treated with GEF-H1 siRNA or control siRNA and infected with MNV for 12 h. RNA samples and supernatants were harvested and analysed via RT-qPCR and plaque assay, respectively (Figure 5C,D). We observed no significant difference in the amount of viral RNA generated during the MNV infection between cells treated with GEF-H1 siRNA, the control siRNA, or the untreated cells (Figure 5C). The amount of viral RNA also correlated with the amount of viral proteins produced under the same conditions (Figure 5A). Additionally, viral titres between cells that were treated with GEF-H1 siRNA, the control siRNA, or left untreated showed no statistical difference (Figure 5D).

This indicates that GEF-H1 does not influence MNV RNA replication, viral protein production, or the assembly or release of progeny infectious MNV particles. Combined, these results put into question the postulated critical role GEF-H1 plays in viral sensing. However, we also cannot discount that MNV is counteracting the GEF-H1 antiviral function.

### 3.5. Suppression of the GEF-H1 Expression Does Not Affect the Production of Cytokines in Response to Poly(I:C) Stimulation or MNV Infection

The knockdown of GEF-H1 in murine macrophages did not lead to a change in MNV replication unlike previous studies performed with influenza A virus (IAV; [25]). The role of GEF-H1 among others has been characterized as an adaptor protein to enhance host viral sensing via the RIG-I/MDA5 pathway. A downstream effect of this signalling pathway is the induction of type I IFNs, mainly IFNβ. To test if the reduction of GEF-H1 in murine macrophages cells affects the induction of IFNβ, we again treated cells with GEF-H1 siRNA, the control siRNA, or left them untreated, before infecting them with MNV for 12 h or treating them with poly(I:C). Poly(I:C) is a dsRNA analogue and was used as a positive control to induce IFNβ transcription. After infection or stimulation, RNA samples were harvested and analysed for the expression of IFNβ mRNA, as well as TNFα mRNA as a control using RT-qPCR (Figure 6).

Poly(I:C)-treated and MNV-infected cells both showed the induction of IFNβ and TNFα compared to uninfected cells (Figure 6A,B, respectively). In the case of IFNβ, poly(I:C) treatment generated lower amounts of IFNβ mRNA compared to MNV-infected cells, whereas the levels of TNFα transcription did not differ significantly between MNV-infected and poly(I:C)-treated cells (Figure 6A,B). We observed that the IFNβ and TNFα mRNA levels did not differ significantly between GEF-H1 siRNA-treated, control siRNA-treated, and untreated cells when cells were infected with MNV (Figure 6C,D). This result suggests that GEF-H1 is not essential for the induction of type I IFN or TNFα in MNV-infected cells.

The data we have obtained so far indicates that GEF-H1 is not essential for the induction of IFNβ or TNFα during MNV infection and that a reduction in GEF-H1 does not promote nor restrict viral replication. These observations stand in stark contrast to studies on IAV highlighting the importance of GEF-H1 to induce type I IFN [25]. That study also showed that the lack of GEF-H1 leads to higher viral titres and viral mRNA levels. To test the impact of GEF-H1 on innate immune sensing, we repeated our GEF-H1 siRNA silencing experiments with influenza strain X31, a mouse specific IAV strain, as well as MNV and the flavivirus West Nile Virus (WNV) as a control. The knockdown as well as control cells were infected with MNV, X31, or WNV and harvested at the virus specific peak replication time points (12 h, 18 h, and 21 h post-infection, respectively). Cells were harvested and analysed for the mRNA levels of IFNβ, TNFα, and viral mRNA via RT-qPCR (Figure 7C,D). We analysed the change in IFNβ, TNFα, and viral mRNA induction by comparing infected siRNA-treated cells (GEF-H1 siRNA and control siRNA) to infected but untreated cells. As expected, there was no significant change in the TNFα mRNA levels for any of the viruses when cells were treated with GEF-H1 or the control siRNA because GEF-H1 has so far not been described to play a role in the TNFα signalling pathway. In our hands, we were unable to detect a difference in the IFNβ mRNA levels in GEF-H1 knockdown cells infected with X31 (Figure 6C). Additionally, we did not observe an increase in X31 viral RNA in the GEF-H1 siRNA-treated and X31-infected cells (Figure 6E). Surprisingly, GEF-H1 siRNA-treated and MNV-infected cells showed a small but significant increase in the levels of viral mRNA compared to the control siRNA-treated cells (Figure 6E).

Overall, we observed that the suppression of GEF-H1 via siRNA did not contribute significantly to the induction of the innate immune response, particularly the induction of IFNβ, nor does it affect the replication kinetics of a range of RNA viruses including both positive-sense (e.g., MNV and WNV) and negative-sense viruses (e.g., influenza virus).

## 4. Discussion

In this study, we aimed to determine the contribution of the host microtubule-associated and innate immune sensing protein GEF-H1 during the replication cycle of MNV. We investigated the association of GEF-H1 with MNV and its contribution to innate immune regulation, where we observed no significance influence of GEF-H1 on immune detection but a small yet significant influence on the control of MNV replication (Figure 6). Intriguingly, we did observe that GEF-H1 showed partial co-localisation with the MNV RC (Figure 1) and a substantial association between MNV NS3 and GEF-H1 (Figure 3), suggesting that during infection, the NS3 protein may modulate the functional capacity of GEF-H1 to sense and respond to MNV infection. However, our observed co-localisation of GEF-H1 with the MNV did display some regions of partial or no overlap within the RC. One speculation of this could be that there is a distinct compartmentalisation of functions within the RC, i.e., separate sites for genome replication vs. translation. Thus, it could be that GEF-H1 plays a very specific role within the RC, although we have not yet elucidated what this may be.

GEF-H1 and the MNV NS3 proteins have both been observed to associate with microtubules [15,19,20]. During our transient transfection studies, we observed that co-expression of the MNV NS3 protein and GEF-H1 significantly affected the morphology and localisation of GEF-H1. GEF-H1 and NS3 co-expressing cells displayed a rather diffuse labelling for GEF-H1 without the formation of microtubule fibres. In addition, when co-expressed with NS3, GEF-H1 no longer appeared to co-localise to the microtubules but displayed a more prominent partial co-localisation with NS3 (Figure 4), suggesting that NS3 may alter the ability of GEF-H1 to associate with microtubules. The association of both proteins was subsequently confirmed via immune precipitation (Figure 3), suggesting a direct interaction between both of these proteins.

Although we have not explored the interaction between NS3 and GEF-H1 in great detail, we hypothesise that NS3 could affect GEF-H1 via different mechanisms (Figure 7): by direct association with GEF-H1, e.g., through binding and therefore inhibiting GEF-H1 or by changing the phosphorylation status of GEF-H1. GEF-H1 is postulated to be inactive and binds to microtubules in its phosphorylated state, whereas dephosphorylation leads to the release from the microtubules and the subsequent activation of GEF-H1 [25]. The hypothesis of a direct interaction would be supported by the findings that co-expression of NS3 seemingly leads to the detachment of GEF-H1 from the microtubules and that GEF-H1 and NS3 co-localise and can be precipitated together (Figure 3).

Another possibility is an indirect interaction of NS3 with GEF-H1 through their common binding partner, the microtubules, and more specifically tubulin. NS3 has been shown to partially co-localise with the microtubule marker β–tubulin, and its mobility in the cells is dependent on the microtubule network [15]. As previously shown, GEF-H1 also co-localises with tubulin and could therefore be affected by changes in the microtubule network. Changes like that could be the result of another protein binding to tubulin, such as NS3. The expression of NS3 and subsequent binding of the protein to the microtubules could interfere with the binding of GEF-H1 to the tubulin and indirectly release GEF-H1 from the microtubules, resulting in a similar phenotype as a direct interaction of both proteins would. However, this hypothesis is contested by the direct association of NS3 and GEF-H1 in the immunoprecipitation assay supporting a direct interaction.

We observed that the expression of GEF-H1 and its mutants lead to changes in the size and number of viral RCs, except for the C53R mutant which is not able to bind to the microtubules. This indicates that removing the ability of GEF-H1 to bind to the microtubules and its potential mediator abolishes the effect that the GEF-H1 expression has on the MNV RC, changing the location and amount of MNV RCs. Our previous studies indicated a role for microtubules in the positioning of the MNV RC juxtaposed to the MTOC [16]. Thus, it would appear that modulation of microtubule dynamics influences the ability of MNV to replicate effectively. In the model proposed by Chiang et al., [25] GEF-H1 aids in viral sensing once it has been released from the microtubules. This release is associated with the dephosphorylation of the protein and a reduction in microtubule binding. Based on this model, we expected to observe a change from the microtubule-like staining of GEF-H1 in the immunofluorescence microscopy analysis; however, we did not observe this change in localisation.

To further disentangle the role of GEF-H1 during MNV replication and innate immune sensing, the GEF-H1 expression was suppressed via a siRNA treatment in mouse macrophages. We observed that the knockdown of GEF-H1 did not reduce the level of IFNβ mRNA produced during an MNV infection, indicating that GEF-H1 was not essential for nucleic acid sensing or activating the innate immune response during infection. In addition, the suppression of GEF-H1 did result in a small but significant increase in viral replication, hinting on the fact, again, that GEF-H1 might not play a critical role in combatting a MNV infection. All these observations indicate that the postulated role of GEF-H1 in the viral sensing pathway might either be specific for other viruses, such as IAV, or not as crucial as it has been hypothesised [25]. To reconcile our results with the published observations, we tested two other viruses, X31 (IAV) and WNV, in our GEF-H1 siRNA knockdown model and similar to MNV, we failed to see an effect of GEF-H1 on the viral replication although we did observe a significant reduction in the induction of IFNβ mRNA during the WNV infection (Figure 6C). To conclusively prove that GEF-H1 does not influence the MNV replication, we would need to deplete the protein expression via CRISPR technology. However, we would also suggest that an approximate 95% reduction in the GEF-H1 expression should also indicate some influence on the MNV replication if it was critical.

Overall, we have observed that the cellular microtubule-associated protein GEF-H1 partially associates with the MNV RC. However, this association between GEF-H1 and MNV RNA replication does not appear to be entirely essential as siRNA treatments specific for GEF-H1 resulted in only a very small increase in MNV replication. However, we did observe that the MNV NS3 protein can associate with GEF-H1 during transient over-expression of both these proteins. Intriguingly, NS3 could affect and negate the microtubule-bundling properties of GEF-H1, indicating that the MNV NS3 protein most likely affects microtubule dynamics and supports previous studies from our lab on the connections between MNV, NS3, and microtubules [15,16].

## Figures and Tables

**Figure 1 viruses-11-00047-f001:**
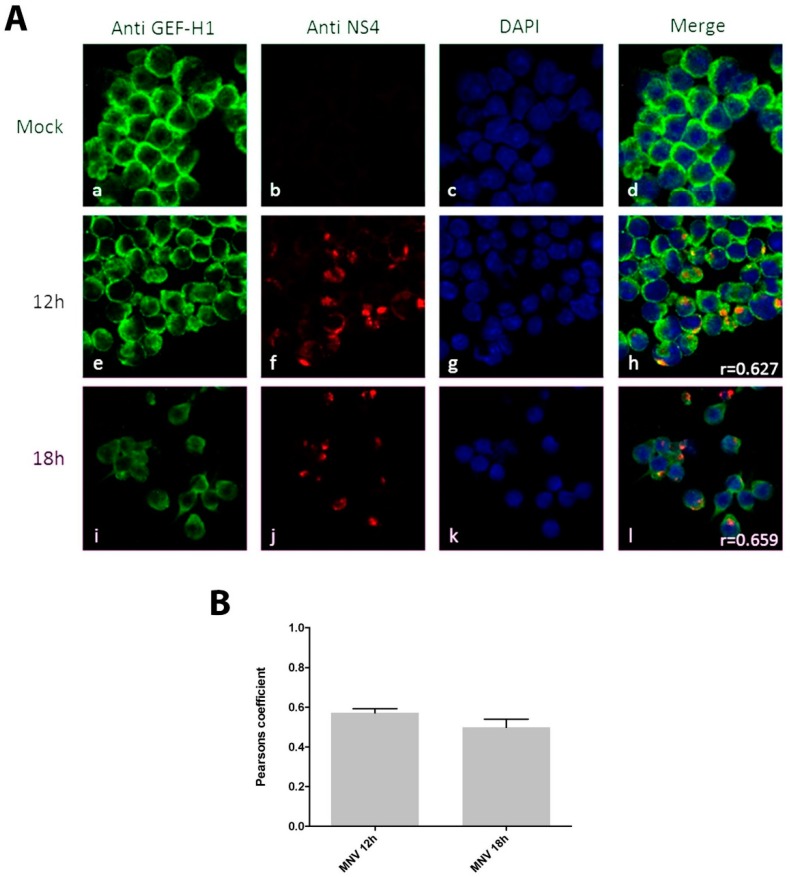
The replication complex (RC) of murine norovirus (MNV) co-localises with the endogenous guanine nucleotide exchange factor-H1 (GEF-H1) in the perinuclear area. (**A**) RAW264.7 cells were infected with MNV (multiplicity of infection (MOI) 5) and fixed at 12 h and 18 h after infection. Panels a, e, and i show the staining for the anti-GEF-H1 antibody (green); panels b, f, and j display the staining with an anti-non-structural protein 4 (NS4) antibody (red); panel c, g, and k represent the nuclear staining with DAPI (blue); and panels d, h, and l show the merged pictured of all channels. Stained cells were analysed via confocal microscopy and the co-localisation was quantified with the Pearson’s coefficient. (**B**) Quantitation of the Pearson’s coefficients. MNV 12 h = 0.57 ± 0.08 (*n* = 16) and MNV 18 h = 0.50 ± 0.16 (*n* = 14). Bars represent average ± standard error of the mean (SEM). Images were collected over triplicate experiments and analysed in GraphPad Prism (www.graphpad.com/scientific-software/prism/).

**Figure 2 viruses-11-00047-f002:**
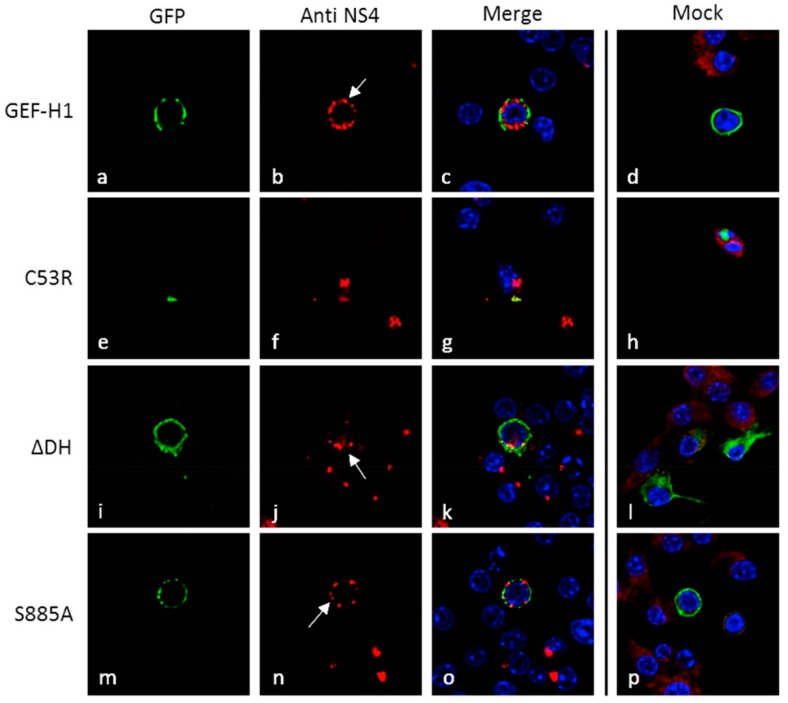
The expression of GEF-H1 WT induces the dispersion of the MNV RC. Murine macrophages (RAW264.7) were transfected with GFP-tagged GEF-H1 forms (WT, C53R: unable to associate with microtubules, ΔDH: no GEF activity, S885A: constantly active GEF) and at 12 h post-transfection (h.p.t.) were subsequently infected with MNV (MOI 5). Cells were fixed at 12 h.p.i. and immune-labelled with an anti-NS4 antibody for confocal microscopy analysis. The GFP signal of the different GEF-H1 forms is displayed in panels a, e, i, and m (green), while panels b, f, j, and n show the staining with the anti-NS4 antibody (red). Panels c, g, k, and o show the merged image including the nuclear stain (DAPI; blue). Panels d, h, l, and p represent the merged image of the GEF-H1-GFP transfected but uninfected cells. Macrophages expressing the microtubule-associated forms of GEF-H1 (WT, ΔDH, S885A) all showed a dispersion of the MNV RC (white arrows).

**Figure 3 viruses-11-00047-f003:**
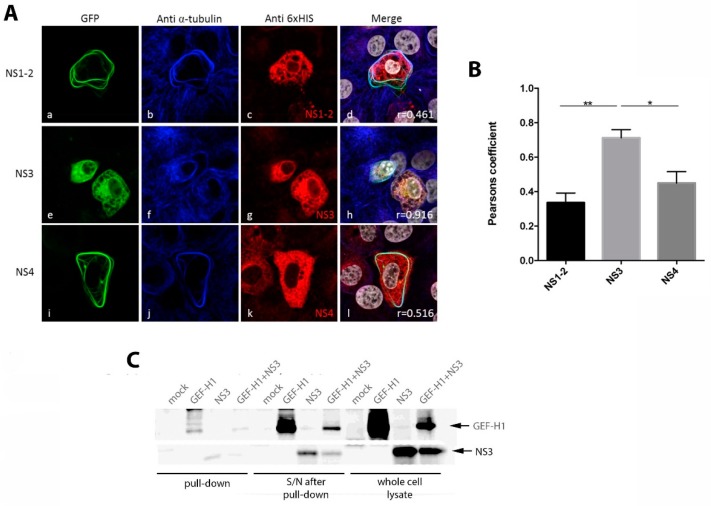
Co-expression with non-structural protein 3 (NS3) changes the GEF-H1 morphology. (**A**) Vero cells were co-transfected with GEF-H1-GFP WT and the HIS-tagged viral proteins. Cells expressing GEF-H1-GFP is shown in panels a, e, and i (green). Panels b, f, and j display the staining with an anti-α–tubulin antibody (blue), while panels c, g, and k show the antibody staining for anti-6× HIS (red). Panels d, h, and l represent the merged image of all channels including the nuclear stain (grey). Co-transfected cells displayed the typical bundle-like structures of GEF-H1 in the cell periphery, except for cells co-transfected with NS3. NS3 and GEF-H1 WT appeared to co-localise and a rather reticular distribution of GEF-H1 could be observed. Additionally, GEF-H1 WT appeared to change the distribution of NS3 as well, compared to cells transfected with NS3 only. The co-localisation was quantified with the Pearson’s coefficient and is indicated in the merged image. (**B**) Quantitation of the Pearson’s coefficients. NS1–2 = 0.34 ± 0.20 (*n* = 9), NS3 = 0.70 ± 0.11 (*n* = 17), and NS4 = 0.45 ± 0.13 (*n* = 9). Bars represent average ± SEM and * = *p* < 0.05 and ** = *p* < 0.01. Images were collected over triplicate experiments and analysed in GraphPad Prism. (**C**) The 293T cells were co-transfected with cDNA expression plasmids encoding for 6× HIS-tagged NS3 and GEF-H1-GFP proteins, and cell lysates were analysed via immune precipitation. Lysates were incubated with a cOmplete^TM^ His-Tag Purification Resin to pull down the 6× HIS-tagged NS3 and possible interaction partners. Lysates were then transferred to a nitrocellulose membrane for immunoblotting with anti-GFP and anti-6× HIS antibodies and visualised by chemiluminescence. A representative image from three independent experiments is presented.

**Figure 4 viruses-11-00047-f004:**
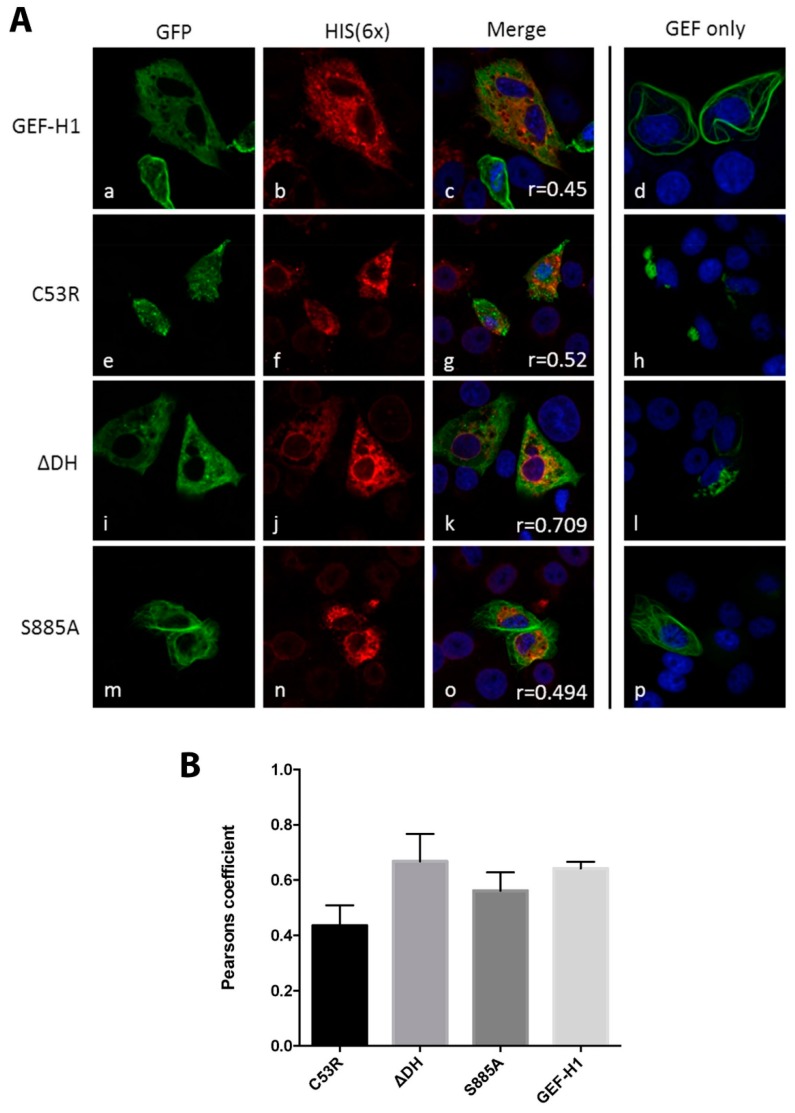
Expression of NS3 changes the location and morphology of the different GEF-H1 forms. (**A**) Vero cells were co-transfected with the HIS-tagged viral protein NS3 and the four different GEF-H1-GFP forms (WT, C53R, ∆DH, S885A). The GFP signal of the different GEF-H1 forms is displayed in panels a, e, I, and m, while panels b, f, j, and n show the staining with the anti-6× HIS antibody. Panels c, g, k, and o show the merged image including the nuclear stain (DAPI). Panels d, h, l, and p represent the merged image of GEF-H1-GFP transfected but uninfected cells. Co-transfected cells displayed a dispersed morphology of GEF-H1 compared to cells transfected with the GEF-H1 forms only (panels d, h, l, and p). The co-localisation was quantified with the Pearson’s coefficient and is indicated in the merged image (panels c, g, k, and o). (**B**) Quantitation of the Pearson’s coefficients: C53R = 0.42 ± 0.19 (*n* = 12), ∆DH = 0.66 ± 0.18 (*n* = 15), S885A = 0.60 ± 0.14 (*n* = 17), and GEF-H1 = 0.65 ± 0.01 (*n* = 19). Bars represent average ± SEM. Images were collected over triplicate experiments and analysed in GraphPad Prism.

**Figure 5 viruses-11-00047-f005:**
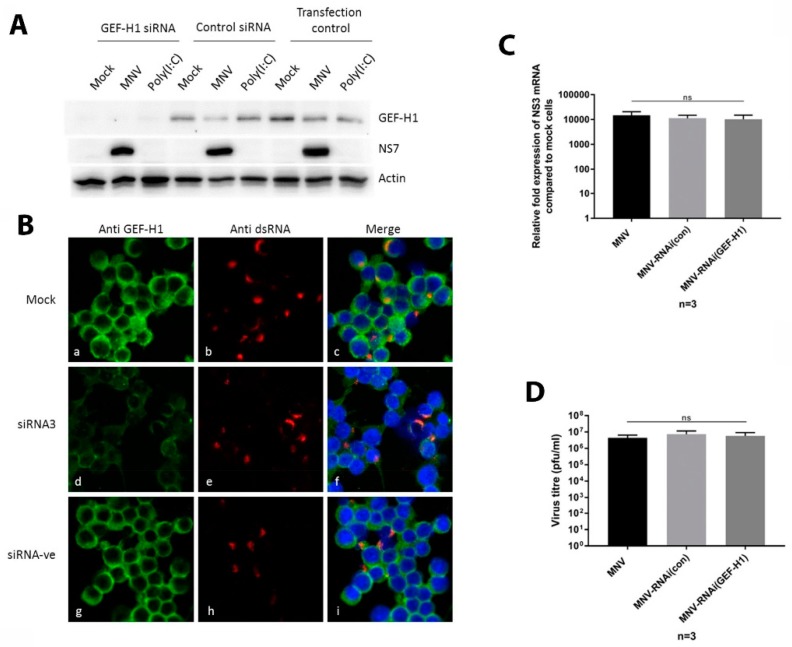
Suppression of the GEF-H1 expression does not influence MNV replication or the protein production of the infectious virus release. RAW264.7 cells were treated twice with GEF-H1 siRNA (siRNA3) or control siRNA (siRNA-ve), before cells were infected with MNV (MOI 5) for 12 h. (**A**) The immunoblot analysis of whole cell lysates stained with antibodies against GEF-H1, NS7, and actin. (**B**) Panels a, d, and g show the staining for the anti-GEF-H1 antibody; panels b, e, and h display the staining with an anti-dsRNA antibody; and panels c, f, and i represent the merged pictured of all the channels including the nuclear stain (DAPI). Stained cells were analysed via confocal microscopy. (**C**) Cells were lysed, and RNA was extracted. The relative fold expression of MNV mRNA compared to mock cells was analysed via RT-qPCR. (**D**) The supernatant of the infected cells was collected and used to determine viral titres via plaque assay. (*n* = 3, average ± SEM, not significant [ns]: *p* > 0.05; one-way ANOVA).

**Figure 6 viruses-11-00047-f006:**
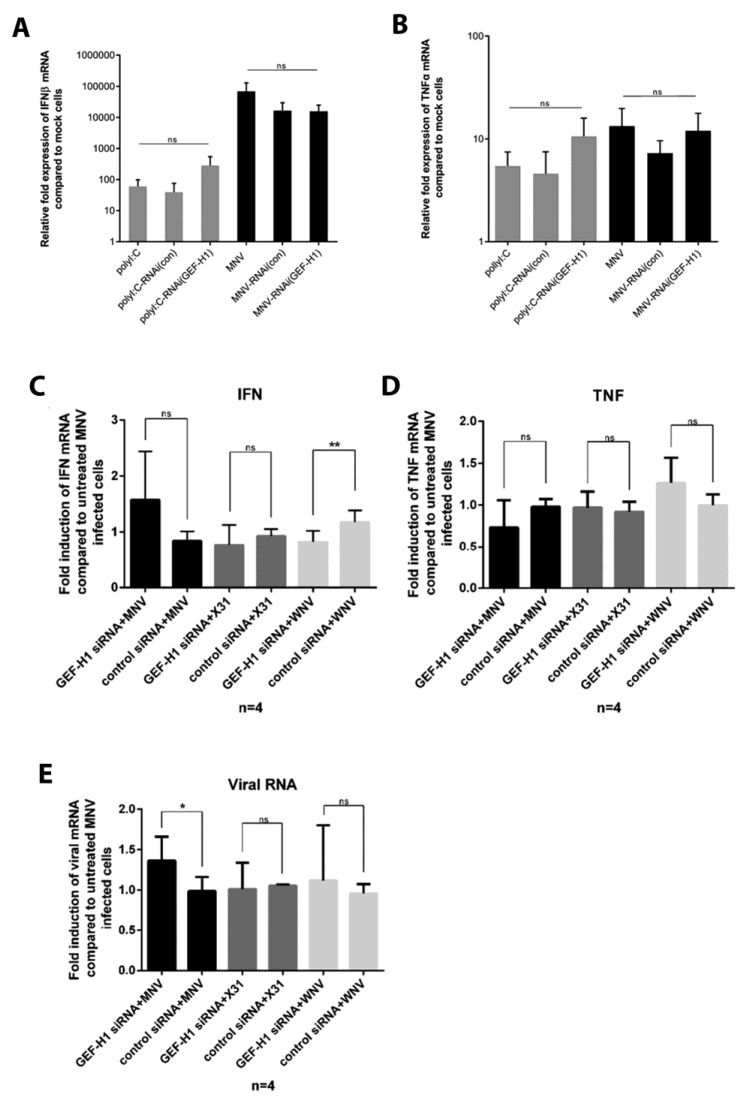
Knockdown of GEF-H1 in murine macrophage cells does not lead to significant changes in the mRNA expression of IFNβ and TNFα upon MNV infection. RAW264.7 cells were treated twice with GEF-H1 siRNA (siRNA3) or the control siRNA (siRNA-ve) before cells were either infected with MNV (MOI 5), treated with poly(I:C), or left untreated for 12 h. Cells were lysed, and the RNA was extracted. Relative fold expression of IFNβ (panel A) and TNFα mRNA (panel B) compared to mock cells was analysed via RT-qPCR (*n* = 3, average +/− SEM, not significant [ns]: *p* > 0.05; one-way ANOVA). In panels C and D, RAW264.7 cells were treated twice with GEF-H1 siRNA, the control siRNA, or left untreated before cells were either infected with MNV (MOI 5), X31 (MOI 5), or West Nile Virus (WNV) (MOI 1). Cells were lysed, and the RNA was extracted. Relative fold expression of IFNβ (**A**,**C**), TNFα (**B**,**D**), and viral mRNA (**E**) compared to infected but untreated cells was analysed via RT-qPCR (*n* = 4, average ± SEM, not significant [ns]: *p* > 0.05, * *p* < 0.05; one-way ANOVA).

**Figure 7 viruses-11-00047-f007:**
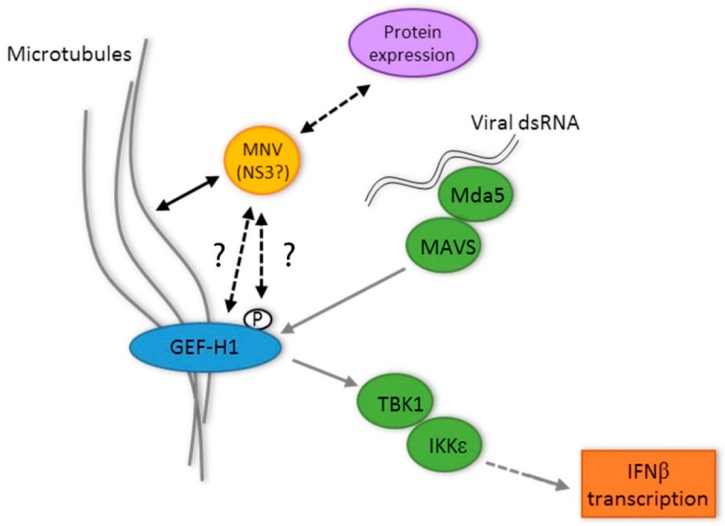
Proposed for the interplay between GEF-H1, MNV replication, microtubules, the MNV NS3 protein, and the innate immune response during the MNV infection of macrophages. Question marks (?) indicate unanswered understanding between the MNV NS3 protein and GEF-H1.

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
