# Peer review of "The Microtubule-Associated Innate Immune Sensor GEF-H1 Does Not Influence Mouse Norovirus Replication in Murine Macrophages"

_viruses, 2019, doi:10.3390/v11010047_

Round 1
Reviewer 1 Report
The manuscript by Fritzlar et al. describes the role of GEF-H1 during murine norovirus infection. In the manuscript the authors deduce that GEF-H1 was bound by MNV NS3, GEF-H1 cellular distribution changed upon NS3 expression, and viral RC were altered when mutant forms of GEF-H1 were expressed. Yet, siRNA knockdown of GEF-H1 did not affect IFN-beta transcript levels, the number of virus-infected cells, and only a small increase in viral RNA titers was observed compared to control siRNA treated cells. Based on these data, the authors conclude that GEF-H1 doesn’t influence MNV replication. However, much of the conclusions are based on immunofluorescence data that shows a single field of view and no quantification across multiple experiments is provided. Without such analysis, it is not possible to make conclusions. In addition, conflicting data is presented when comparing figures as outlined below in detail. Hence, in its current form the manuscript is preliminary.
Major comments
1) The authors make several conclusions regarding the colocalization of proteins based on immunofluorescence data and Person’s coefficient. However, data shown in Figures 1, 2, 3, 4, 5A-B only show one field of view from experiment. No quantification across multiple experiments is shown. This is critical to clarify how reproducible the data is. For example, The Pearson coefficient between GEF-H1and NS3 in Figure 3A(h) is 0.916, while in Figure 4(c) it is 0.45, even though this is the same experiment. This suggests that the spread of the data may be quite large, reducing the ability to make conclusions.
2) The authors conclude that GEF-H1 does not affect innate immune signaling and virus infection based on work in RAW cells. However, many transformed cells have defects in innate immune signaling (doi: 10.1007/978-1-61779-340-0_15). Thus, it would be important to repeat these experiments in primary murine macrophages.
3) In Figure 6E the authors see a significant increase in viral genome titers in GEF-H1 siRNA-treated cells vs control siRNA cells. This contradicts data shown in Figure 5C, when no difference was seen between those two treatments looking at NS3 mRNA. Since part of the viral genome contains NS3, why are the results divergent?
Minor comments
1) Page 5, line 5: CW1 is not a MNV strain but instead a plaque isolate of MNV-1. Thus, the strain is MNV-1. Please correct the nomenclature.
2) Page 16, 5th line in discussion: The authors state: “no significant influence of GEF-H1 on … control of MNV replication”. However, a statistically significant difference was observed in Figure 6E. Thus, the conclusion (and the title, which implies a similar conclusion) should be modified.
Author Response
Reviewer 1
Major comments
1) The authors make several conclusions regarding the colocalization of proteins based on immunofluorescence data and Person’s coefficient. However, data shown in Figures 1, 2, 3, 4, 5A-B only show one field of view from experiment. No quantification across multiple experiments is shown. This is critical to clarify how reproducible the data is. For example, The Pearson coefficient between GEF-H1and NS3 in Figure 3A(h) is 0.916, while in Figure 4(c) it is 0.45, even though this is the same experiment. This suggests that the spread of the data may be quite large, reducing the ability to make conclusions.
- We have now provided quantitation of data cross triplicate experiments for the IF analyses described in Figs 1, 3 and 4. We have not included such data for Fig 2 as we have concluded that minimal co-localisation exists
2) The authors conclude that GEF-H1 does not affect innate immune signaling and virus infection based on work in RAW cells. However, many transformed cells have defects in innate immune signaling (doi: 10.1007/978-1-61779-340-0_15). Thus, it would be important to repeat these experiments in primary murine macrophages.
- The use of primary macrophages would be a worthy experiment and one that we have discussed. However, infection rates in primary macrophages (in our hands) is between 1-5% and transfection rates are equally as poor. Thus, we do not envisage that such studies would be beneficial. We are also aware of the deficiencies of RAW cells however, these deficiencies are for inflammasome and DNA sensing and as such would not influences our studies. We have shown that RAW are effectively activated upon MNV infection (see Enosi Tuipulotu, et al, (2017). RNA Sequencing of Murine Norovirus-Infected Cells Reveals Transcriptional Alteration of Genes Important to Viral Recognition and Antigen Presentation. Front Immunol 8:959).
3) In Figure 6E the authors see a significant increase in viral genome titers in GEF-H1 siRNA-treated cells vs control siRNA cells. This contradicts data shown in Figure 5C, when no difference was seen between those two treatments looking at NS3 mRNA. Since part of the viral genome contains NS3, why are the results divergent?
- I believe the reviewer may have mis-interpreted this Figure. Fig 5C depicts MNV viral RNA compared to mock cells whereas Fig 6E depicts MNV viral RNA (as a fold change) to untreated MNV-infected cells. Thus there is a major difference between the two experiments.
Minor comments
1) Page 5, line 5: CW1 is not a MNV strain but instead a plaque isolate of MNV-1. Thus, the strain is MNV-1. Please correct the nomenclature.
- We received the MNV-1 (CW1) isolated from the Virgin lab many years ago and I believe it is useful to mention that the virus we use is the original isolate and not one adapted by passage in cell culture.
2) Page 16, 5th line in discussion: The authors state: “no significant influence of GEF-H1 on … control of MNV replication”. However, a statistically significant difference was observed in Figure 6E. Thus, the conclusion (and the title, which implies a similar conclusion) should be modified.
- Thank you and we have adjusted the text accordingly
Reviewer 2 Report
Fritzlar et al. examined the effect of guanine nucleotide exchange factor-H1 (GEF-H1) on murine norovirus (MNV) replication. GEF-H1 was previously reported to play a role in viral RNA sensing and IRF3 dependent interferon (IFN) induction.
The authors describe an association of GEF-H1 with viral replication complexes and the viral protein NS3 specifically. They report that suppression of endogenous GEF-H1 with siRNA did not affect MNV replication but that overexpression of a GFP-GEF-H1 severely changes MNV replication complex morphology. The authors also challenge the previously reported finding that GEF-H1 contributes to IFN induction.
There are a number of areas in which I believe the manuscript could be improved.
Figure 1. The authors conclude that GEF-H1 ‘partially’ co-localised with the MNV replication complex. In the green channel (anti-GEF-H1) id dispersed throughout the cytoplasm. At 12 and 18 hrs post infection there is less green signal overall but it still seems to be widely distributed, perhaps accounting for the ‘partial’ overlap. In contrast the replication complexes (red channel) are tightly formed and in the overlap remain largely red. Are the authors suggesting that the replication complex is not homogenous and that GEF-H1 only interacts with specific regions of it? The overall even intensity of the signal for the replication complex wold argue against this. The authors should provide an explanation for this partial overlap and explain how it fits with their conclusions.
Figure 2. Why does the GFP-GEF-H1 signal not overlap with the replication complex, even partially, if the authors conclusions are correct? Does the GFP-tagged protein not behave in a similar way to the endogenous protein? If not, then is it a suitable tool? How does the level of overexpressed protein compare to the endogenous levels? In panels d, h, l, and p, where cells are transfected with GFP-GEF-H1, but not infected, what is the red signal, which in the other panels corresponds to NS4 and the viral replication complex?
Figure 3B. What is the signal present in the mock (Precipitate) and NS3-His (Lysate w/o precipitate) anti-GFP gel? This is ignored by the authors. The full gel should be shown, if not in the main figure, then in the supplemental information for clarity. In the anti-His blot, why is the anti-His (NS3) signal so much stronger than in the cells co-expressing the GEF-H1-GFP than not? Does co-expression increase expression of NS3? The authors do not actually show what NS3 overexpression looks like on its own, which may help account for this. Instead they state that that the localisation of NS3 is altered by co-expression with GEF-H1-GFP, and reference a previous paper. This is not appropriate; the authors should perform these single transfections alongside the double expressions to act as suitable controls. This data should be presented, again, in the main figure or as supplemental information. Why was a loading control for the western not included, at least for the lysate w/o precipitate samples? How many times was this experiment performed, are these representative gels? Some indication of reproducibility should be provided.
Figure 5. The siRNA KD is incomplete as shown by the IF data in panel B. This could account for the differences between the authors subsequent data in figure 6 and that previously published by Chiang et al., Nature Immunology 2014, in which macrophages from GEF-H1 knock out mice (completely gone) were used. I was also surprised that the authors did not discuss the work by Wang et al., Cell Cycle, 2017, in which GEF-H1 depletion decreased IFN stimulation in mycobacterium infected macrophages. I appreciate that this is of course a different infection system, but the results broadly match those of Chiang et al. and not those presented here. Can the authors obtain the knock out mouse cells and see if the data is reproducible in their hands or perhaps use a system to knock out instead of knock down to completely remove GEF-H1 and test again? If the effect of GEF-H1 is not reproducible then this is of course of major importance for the field but the data presented here has not yet demonstrated that this is the case.
Figure 6E. With such large error bars, it is difficult to appreciate the importance of assigning significance to the MNV data versus the WNV data.
Figure 7. The authors have presented data to challenge the previous findings that GEF-H1 acts in IFN induction and by their own admission have done little to characterise their proposed finding that GEF-H1 interacts directly with the replication complex and NS3. Therefore, this model could be modified as to have more question marks between NS3 and GEF-H1. As it is the model is not very useful. Also, compared to the detailed figure legends provided elsewhere, there is nothing in Figure 7 to help the reader.
Minor comments.
Remove lines 44 and 45 from the introduction.
Abbreviate replication complex in the introduction.
Introduction line 59, delete ‘observed to be’.
Introduction line 60, please replace ‘be responsible for’ with ‘contribute to’.
Introduction lines 77-81, please rephrase.
In Figure 1, there are no panels n, o or p, please remove from the legend.
In Figure 5C, please replace ‘NS3 mRNA’ with ‘viral RNA’ in the axis legend.
Reviewer 3 Report
AUTHORS
Manuscript ID : viruses-402800
Title : The microtubule-associated innate immune sensor GEF-H1 does not influence Mouse Norovirus replication in murine macrophages
This manuscript is very interesting, providing novel insights on MNV replication. It is well written in good English language and it will likely have an impact in its field. I have a few questions and comments:
Please describe acronym RC (replication complex) in abstract
Did authors design qPCR primers or used previously designed ones? If the later, please reference.
Why did authors describe RT step conditions and not qPCR ones? Please describe these conditions as well.
Figure 1 caption refers to panels m, n and p but cannot see these?
Figure 5 has 2 plots (bars), with “NS”. Please put on caption the meaning of “NS”. If it is Not Significant, please describe which test was used to conclude no statistically significant differences were observed. Same for figure 6
Author Response
Reviewer 3
I have a few questions and comments:
Please describe acronym RC (replication complex) in abstract
- done
Did authors design qPCR primers or used previously designed ones? If the later, please reference. Why did authors describe RT step conditions and not qPCR ones? Please describe these conditions as well.
- Yes these have been described before and we have now cited the reference. The primer sequences and details can be obtained from the authors as indicated
Figure 1 caption refers to panels m, n and p but cannot see these?
- Thank you these have been removed
Figure 5 has 2 plots (bars), with “NS”. Please put on caption the meaning of “NS”. If it is Not Significant, please describe which test was used to conclude no statistically significant differences were observed. Same for figure 6
- done
Round 2
Reviewer 1 Report
The authors nicely address the previous critiques. I have no further comments.
Author Response
Thank you
Reviewer 2 Report
1. Figure 1. The authors conclude that GEF-H1 ‘partially’ co-localised with the MNV replication complex. In the green channel (anti-GEF-H1) id dispersed throughout the cytoplasm. At 12 and 18 hrs post infection there is less green signal overall but it still seems to be widely distributed, perhaps accounting for the ‘partial’ overlap. In contrast the replication complexes (red channel) are tightly formed and in the overlap remain largely red. Are the authors suggesting that the replication complex is not homogenous and that GEF-H1 only interacts with specific regions of it? The overall even intensity of the signal for the replication complex wold argue against this. The authors should provide an explanation for this partial overlap and explain how it fits with their conclusions.
Response
Apologies this may have been the wrong term to use, we are suggesting that some of the GEF-H1 pool is co-localising within the MNV RC. We have now included quantitation of the Pearson’s coefficients to support our observations.
Thank you for including the Pearson's coefficients throughout which are useful. However there is still no discussion on why there may be large regions of no overlap within a single RC.
2. Figure 2. Why does the GFP-GEF-H1 signal not overlap with the replication complex, even partially, if the authors conclusions are correct?
Response
As described within the text we observed that expression of the different GEF-H1 mutants actually affected the formation and distribution of the complex, most likely via effecting microtubule dynamics. Thus there is a distinct and disparate staining pattern for the MNV RC and the GEF-H1 mutants
In Figure 1 there is some overlap between endogenous GEF-H1 and the replication complex. In Figure 2 there is no overlap between the GFP-WT-GEF-H1, or the mutants, and the dispersed RC. That suggests that there is something missing or altered in the RCs that GFP-GEF-H1 can no longer interact with, perhaps caused by over expression and messing up of microtubules, or that the GFP-GEF-H1 cannot interact with the RC in the same way as the endogenous protein.
3. Does the GFP-tagged protein not behave in a similar way to the endogenous protein? If not, then is it a suitable tool? How does the level of overexpressed protein compare to the endogenous levels?
Response
The WT GEF-H1 displays similar properties to the endogenous GEF-H1 but there is more associated with the cytoskeleton at the periphery of the cell. We are unsure if the mutants behave the same as this is the only way to analyse and observed them. These mutant constructs can only be expressed from recombinant plasmids so they are the only suitable tools for these analyses. The expression of the GEF-H1 from a recombinant plasmid does produce a lot more protein as the cellular vs CMV promoters are vastly different
See response to point 2
4. In panels d, h, l, and p, where cells are transfected with GFP-GEF-H1, but not infected, what is the red signal, which in the other panels corresponds to NS4 and the viral replication complex?
Response
Yes transfected but not infected, the red signal is the background staining of the anti- NS4 antibody
Were different conditions captured at different intensities then as there appears to be no equivalent background in panels b-n and c- o. This should be stated.
5. Figure 3B. What is the signal present in the mock (Precipitate) and NS3-His
(Lysate w/o precipitate) anti-GFP gel? This is ignored by the authors. The full gel
should be shown, if not in the main figure, then in the supplemental information for
clarity.
Response
It is a background band observed using the antibody, although we have a suspicion that it may actually be endogenous GEF-H1 as the antibody used in the western blot is an anti-GEF-H1 antibody. However, we have now replaced this blot with a more clearer one. Also note that the membrane was probed with both anti-His and anti- GEF-H1. We have provided the full gel for the reviewer.
This is now Figure 3C. In the original and revised draft the legend states that the blot was probed with anti-GFP and anti-6XHis. Also in the first draft the figure was labelled as GEF-H1-GFP and now is labelled GEF-H1. Which was used? If it is anti-GEF-H1 as in the original blot that therefore suggest that the endogenous GEF-H1 (as it indeed likely is if anti-GEF-H1 ab was used) precipitates with the pull down beads in the absence of His-NS3. If it was instead probed with anti-GFP then it is unlikely to be endogenous GEF-H1 as the authors suggest as the size of the endogenous and tagged proteins would be different. Which antibody was used should be clarified in the legend and the associated text. In the new gel (thank you for providing the whole image) the signal for GEF-H1 after pull down appears as intense in cells co-transfected with HisNS3 as those with only GEF-H1 transfected. Again this suggests that GEF-H1 is precipitating non-specifically with the beads and does not demonstrate an interaction with NS3. The text should be amended accordingly.
6. In the anti-His blot, why is the anti-His (NS3) signal so much stronger than in the
cells co-expressing the GEF-H1-GFP than not? Does co-expression increase
expression of NS3?
Response
There is quite a variation in the degree of expression in these experiments and in fact we generally observe less protein produced upon co-transfection of any plasmid with the NS3 construct. We have currently submitted an additional manuscript that explores this NS3-induced manipulation on translation
I look forward to reading this work.
8. The authors do not actually show what NS3 overexpression looks like on its own,
which may help account for this. Instead they state that that the localisation of NS3 is
altered by co-expression with GEF-H1-GFP, and reference a previous paper. This is
not appropriate; the authors should perform these single transfections alongside the
double expressions to act as suitable controls. This data should be presented, again,
in the main figure or as supplemental information.
Response
Actually the membrane has been probed with antibodies against both His and GEF- H1 and the whole cell lysates show the differences in single vs co-transfection for both NS3 and GEF-H1.
I apologise for my lack of clarity. I meant the authors do not show images of NS3 distribution when over expressed on its own. This would be useful for comparison withe the current data.
9. Why was a loading control for the western not included, at least for the lysate w/o
precipitate samples? How many times was this experiment performed, are these
representative gels? Some indication of reproducibility should be provided.
Response
This experiment was repeated at least 3 times. We have mentioned this in the legend to Figure 3
Thank you for now including this information
10. Figure 5. The siRNA KD is incomplete as shown by the IF data in panel B. This could account for the differences between the authors subsequent data in figure 6 and that previously published by Chiang et al., Nature Immunology 2014, in which macrophages from GEF-H1 knock out mice (completely gone) were used. I was also surprised that the authors did not discuss the work by Wang et al., Cell Cycle, 2017, in which GEF-H1 depletion decreased IFN stimulation in mycobacterium infected macrophages. I appreciate that this is of course a different infection system, but the results broadly match those of Chiang et al. and not those presented here. Can the authors obtain the knock out mouse cells and see if the data is reproducible in their hands or perhaps use a system to knock out instead of knock down to completely remove GEF-H1 and test again? If the effect of GEF-H1 is not reproducible then this is of course of major importance for the field but the data presented here has not yet demonstrated that this is the case.
Response
Thank you for identifying the work by Wang and colleagues, this was an unfortunate omission on our behalf which we have now included. We actually believe that or RNAi repression is excellent based on the western blot data where practically no GEF-H1 can be observed. We cannot fully understand why in our hands suppression of GEF-H1 does not affect cytokine induction as previously described. One major difference in the Wang paper is that GEF-H1 is upregulated during infection thus serving as a true sensor. AS we do not observe this for MNV it would argue the role of GEF-H1 may in fact be pathogen specific. We have thus adjusted our comments accordingly. Unfortunately we do not have access to a GEF- H1 KO mouse to extend our further studies to investigate a GEF-H1 KO.
Based on the WB the KD is strong but not complete and there is a signal in cells by IF. CRISPR technology works in RAWs and would knock out any expression proving the authors findings conclusively.
11. Figure 6E. With such large error bars, it is difficult to appreciate the importance of assigning significance to the MNV data versus the WNV data.
Response
Ok comment noted
12. Figure 7. The authors have presented data to challenge the previous findings that GEF-H1 acts in IFN induction and by their own admission have done little to characterise their proposed finding that GEF-H1 interacts directly with the replication complex and NS3. Therefore, this model could be modified as to have more question marks between NS3 and GEF-H1. As it is the model is not very useful. Also, compared to the detailed figure legends provided elsewhere, there is nothing in Figure 7 to help the reader.
Response
Ok comment noted
The figure and legend are unchanged.
13. Minor comments.
Remove lines 44 and 45 from the introduction.
Response ?? Reason?
This is a research article. Such text is extraneous.
Abbreviate replication complex in the introduction.
Response
It is?
Line 57 of the revised text. It is not.
Introduction line 59, delete ‘observed to be’.
Response
Done
Introduction line 60, please replace ‘be responsible for’ with ‘contribute to’.
Response
Done
Introduction lines 77-81, please rephrase.
Response
Done
In Figure 1, there are no panels n, o or p, please remove from the legend.
Response
Done
In Figure 5C, please replace ‘NS3 mRNA’ with ‘viral RNA’ in the axis legend.
Response
As there is both genomic and subgenomic RNA produced during infection we would rather depict the specificity of the PCR target
Thank you for clarifying.
Author Response
1. Figure 1. The authors conclude that GEF-H1 ‘partially’ co-localised with the MNV replication complex. In the green channel (anti-GEF-H1) id dispersed throughout the cytoplasm. At 12 and 18 hrs post infection there is less green signal overall but it still seems to be widely distributed, perhaps accounting for the ‘partial’ overlap. In contrast the replication complexes (red channel) are tightly formed and in the overlap remain largely red. Are the authors suggesting that the replication complex is not homogenous and that GEF-H1 only interacts with specific regions of it? The overall even intensity of the signal for the replication complex wold argue against this. The authors should provide an explanation for this partial overlap and explain how it fits with their conclusions.
Response
Apologies this may have been the wrong term to use, we are suggesting that some of the GEF-H1 pool is co-localising within the MNV RC. We have now included quantitation of the Pearson’s coefficients to support our observations.
Thank you for including the Pearson's coefficients throughout which are useful. However there is still no discussion on why there may be large regions of no overlap within a single RC.
- We have now included a comment of the possibility of compartmentalisation of function within the RC
2. Figure 2. Why does the GFP-GEF-H1 signal not overlap with the replication complex, even partially, if the authors conclusions are correct?
Response
As described within the text we observed that expression of the different GEF-H1 mutants actually affected the formation and distribution of the complex, most likely via effecting microtubule dynamics. Thus there is a distinct and disparate staining pattern for the MNV RC and the GEF-H1 mutants
In Figure 1 there is some overlap between endogenous GEF-H1 and the replication complex. In Figure 2 there is no overlap between the GFP-WT-GEF-H1, or the mutants, and the dispersed RC. That suggests that there is something missing or altered in the RCs that GFP-GEF-H1 can no longer interact with, perhaps caused by over expression and messing up of microtubules, or that the GFP-GEF-H1 cannot interact with the RC in the same way as the endogenous protein.
- Yes we agree that the over-expression of GEF-H1 most likely influences and alters microtubule dynamics. We have attempted to discuss this in the 2nd, 3rd and 4th paragraphs of the Discussion
3. Does the GFP-tagged protein not behave in a similar way to the endogenous protein? If not, then is it a suitable tool? How does the level of overexpressed protein compare to the endogenous levels?
Response
The WT GEF-H1 displays similar properties to the endogenous GEF-H1 but there is more associated with the cytoskeleton at the periphery of the cell. We are unsure if the mutants behave the same as this is the only way to analyse and observed them. These mutant constructs can only be expressed from recombinant plasmids so they are the only suitable tools for these analyses. The expression of the GEF-H1 from a recombinant plasmid does produce a lot more protein as the cellular vs CMV promoters are vastly different
See response to point 2
4. In panels d, h, l, and p, where cells are transfected with GFP-GEF-H1, but not infected, what is the red signal, which in the other panels corresponds to NS4 and the viral replication complex?
Response
Yes transfected but not infected, the red signal is the background staining of the anti- NS4 antibody
Were different conditions captured at different intensities then as there appears to be no equivalent background in panels b-n and c- o. This should be stated.
- No the same conditions were used to capture all conditions, thus is just variability in the experiments. We do observe this when investigating murine macrophages. We have mentioned that all conditions were equal in the Materials and Methods
5. Figure 3B. What is the signal present in the mock (Precipitate) and NS3-His (Lysate w/o precipitate) anti-GFP gel? This is ignored by the authors. The full gel should be shown, if not in the main figure, then in the supplemental information for clarity.
Response
It is a background band observed using the antibody, although we have a suspicion that it may actually be endogenous GEF-H1 as the antibody used in the western blot is an anti-GEF-H1 antibody. However, we have now replaced this blot with a more clearer one. Also note that the membrane was probed with both anti-His and anti- GEF-H1. We have provided the full gel for the reviewer.
This is now Figure 3C. In the original and revised draft the legend states that the blot was probed with anti-GFP and anti-6XHis. Also in the first draft the figure was labelled as GEF-H1-GFP and now is labelled GEF-H1. Which was used? If it is anti-GEF-H1 as in the original blot that therefore suggest that the endogenous GEF-H1 (as it indeed likely is if anti-GEF-H1 ab was used) precipitates with the pull down beads in the absence of His-NS3. If it was instead probed with anti-GFP then it is unlikely to be endogenous GEF-H1 as the authors suggest as the size of the endogenous and tagged proteins would be different. Which antibody was used should be clarified in the legend and the associated text. In the new gel (thank you for providing the whole image) the signal for GEF-H1 after pull down appears as intense in cells co-transfected with HisNS3 as those with only GEF-H1 transfected. Again this suggests that GEF-H1 is precipitating non-specifically with the beads and does not demonstrate an interaction with NS3. The text should be amended accordingly.
6. In the anti-His blot, why is the anti-His (NS3) signal so much stronger than in the cells co-expressing the GEF-H1-GFP than not? Does co-expression increase expression of NS3?
Response
There is quite a variation in the degree of expression in these experiments and in fact we generally observe less protein produced upon co-transfection of any plasmid with the NS3 construct. We have currently submitted an additional manuscript that explores this NS3-induced manipulation on translation
I look forward to reading this work.
8. The authors do not actually show what NS3 overexpression looks like on its own, which may help account for this. Instead they state that that the localisation of NS3 is altered by co-expression with GEF-H1-GFP, and reference a previous paper. This is not appropriate; the authors should perform these single transfections alongside the double expressions to act as suitable controls. This data should be presented, again, in the main figure or as supplemental information.
Response
Actually the membrane has been probed with antibodies against both His and GEF- H1 and the whole cell lysates show the differences in single vs co-transfection for both NS3 and GEF-H1.
I apologise for my lack of clarity. I meant the authors do not show images of NS3 distribution when over expressed on its own. This would be useful for comparison withe the current data.
- We have now included this as Supplementary Fig 2
9. Why was a loading control for the western not included, at least for the lysate w/o precipitate samples? How many times was this experiment performed, are these representative gels? Some indication of reproducibility should be provided.
Response
This experiment was repeated at least 3 times. We have mentioned this in the legend to Figure 3
Thank you for now including this information
10. Figure 5. The siRNA KD is incomplete as shown by the IF data in panel B. This could account for the differences between the authors subsequent data in figure 6 and that previously published by Chiang et al., Nature Immunology 2014, in which macrophages from GEF-H1 knock out mice (completely gone) were used. I was also surprised that the authors did not discuss the work by Wang et al., Cell Cycle, 2017, in which GEF-H1 depletion decreased IFN stimulation in mycobacterium infected macrophages. I appreciate that this is of course a different infection system, but the results broadly match those of Chiang et al. and not those presented here. Can the authors obtain the knock out mouse cells and see if the data is reproducible in their hands or perhaps use a system to knock out instead of knock down to completely remove GEF-H1 and test again? If the effect of GEF-H1 is not reproducible then this is of course of major importance for the field but the data presented here has not yet demonstrated that this is the case.
Response
Thank you for identifying the work by Wang and colleagues, this was an unfortunate omission on our behalf which we have now included. We actually believe that or RNAi repression is excellent based on the western blot data where practically no GEF-H1 can be observed. We cannot fully understand why in our hands suppression of GEF-H1 does not affect cytokine induction as previously described. One major difference in the Wang paper is that GEF-H1 is upregulated during infection thus serving as a true sensor. AS we do not observe this for MNV it would argue the role of GEF-H1 may in fact be pathogen specific. We have thus adjusted our comments accordingly. Unfortunately we do not have access to a GEF- H1 KO mouse to extend our further studies to investigate a GEF-H1 KO.
Based on the WB the KD is strong but not complete and there is a signal in cells by IF. CRISPR technology works in RAWs and would knock out any expression proving the authors findings conclusively.
- We agree that this approach would be the most conclusive approach, however we would also suggest that a 95% reduction in GEF-H1 expression should also indicate some influence on MNV replication if it was critical. Thus, we have opted not to invest in the CRISPR approach to deplete GEF-H1. We have added a not similar to this in the Discussion.
11. Figure 6E. With such large error bars, it is difficult to appreciate the importance of assigning significance to the MNV data versus the WNV data.
Response
Ok comment noted
12. Figure 7. The authors have presented data to challenge the previous findings that GEF-H1 acts in IFN induction and by their own admission have done little to characterise their proposed finding that GEF-H1 interacts directly with the replication complex and NS3. Therefore, this model could be modified as to have more question marks between NS3 and GEF-H1. As it is the model is not very useful. Also, compared to the detailed figure legends provided elsewhere, there is nothing in Figure 7 to help the reader.
Response
Ok comment noted
The figure and legend are unchanged.
- Now included
13. Minor comments.
Remove lines 44 and 45 from the introduction.
Response ?? Reason?
This is a research article. Such text is extraneous.
- I believe we have now removed this text, if the reviewer is referring to the below statement: “GEF-H1 is released from microtubules and can act on TBK1 which in turn activates the regulatory kinase IKKε. IKKε phosphorylates the transcription factor IRF3, which translocates into the nucleus to promote IFNβ mRNA transcription”.
Abbreviate replication complex in the introduction.
Response
It is?
Line 57 of the revised text. It is not.
- This has described and abbreviated early in the Introduction in the paragraph starting “Our laboratory has been instrumental …”
Introduction line 59, delete ‘observed to be’.
Response
Done
Introduction line 60, please replace ‘be responsible for’ with ‘contribute to’.
Response
Done
Introduction lines 77-81, please rephrase.
Response
Done
In Figure 1, there are no panels n, o or p, please remove from the legend.
Response
Done
In Figure 5C, please replace ‘NS3 mRNA’ with ‘viral RNA’ in the axis legend.
Response
As there is both genomic and subgenomic RNA produced during infection we would rather depict the specificity of the PCR target
Thank you for clarifying.